# Treatment Strategies Guided by [18F]FDG-PET/CT in Patients with Locally Advanced Cervical Cancer and [18F]FDG-Positive Lymph Nodes

**DOI:** 10.3390/cancers16040717

**Published:** 2024-02-08

**Authors:** Ester P. Olthof, Hans H. B. Wenzel, Jacobus van der Velden, Lukas J. A. Stalpers, Constantijne H. Mom, Maaike A. van der Aa

**Affiliations:** 1Department of Research & Development, Netherlands Comprehensive Cancer Organisation, 3511 LC Utrecht, The Netherlands; h.wenzel@iknl.nl (H.H.B.W.); m.vanderaa@iknl.nl (M.A.v.d.A.); 2Centre for Gynaecologic Oncology Amsterdam (CGOA), Department of Gynaecological Oncology, Amsterdam University Medical Centre, 1081 HV Amsterdam, The Netherlands; j.vandervelden@amsterdamumc.nl (J.v.d.V.); c.mom@amsterdamumc.nl (C.H.M.); 3Department of Radiation Oncology, Amsterdam University Medical Centre, 1055 AZ Amsterdam, The Netherlands; l.stalpers@amsterdamumc.nl

**Keywords:** uterine cervical cancer, locally advanced stage, 18FDG, PET/CT, lymphatic metastases, boost, extended-field radiotherapy, debulking

## Abstract

**Simple Summary:**

Current guidelines recommend treatment planning using [^18^F]FDG-PET/CT for the management of advanced cervical cancer, where suspicious lymph nodes may be treated with nodal boosting, extended-field radiotherapy, and/or debulking in addition to standard chemoradiotherapy to improve survival. However, caution must be exercised because of the risk of unnecessary therapy-related toxicity due to the overtreatment of false-positive nodes. Despite this daily dilemma in clinical practice, only a few studies have evaluated the management of [^18^F]FDG-positive nodes. Therefore, this study aimed to assess how often [^18^F]FDG-PET/CT lymph node information is used in the management of advanced-stage cervical cancer. We found that a total of 380/434 patients (88%) received interventions targeting [^18^F]FDG-positive nodes, with the following distribution: nodal boosting (84%), extended-field radiotherapy (78%), and debulking (12%). Despite existing guidelines advocating [^18^F]FDG-PET/CT-guided treatment planning for the management of advanced cervical cancer, this study highlights that not all cases of [^18^F]FDG-positive nodes received an intervention.

**Abstract:**

Background: Modern treatment guidelines for women with advanced cervical cancer recommend staging using 2-deoxy-2-[^18^F]fluoro-D-glucose positron emission computed tomography ([^18^F]FDG-PET/CT). However, the risk of false-positive nodes and therapy-related adverse events requires caution in treatment planning. Using data from the Netherlands Cancer Registry (NCR), we estimated the impact of [^18^F]FDG-PET/CT on treatment management in women with locally advanced cervical cancer, i.e., on nodal boosting, field extension, and/or debulking in cases of suspected lymph nodes. Methods: Women diagnosed between 2009 and 2017, who received chemoradiotherapy for International Federation of Gynaecology and Obstetrics (2009) stage IB2, IIA2-IVB cervical cancer with an [^18^F]FDG-positive node, were retrospectively selected from the NCR database. Patients with pathological nodal examination before treatment were excluded. The frequency of nodal boosting, extended-field radiotherapy, and debulking procedures applied to patients with [^18^F]FDG-positive lymph nodes was evaluated. Results: Among the 434 eligible patients with [^18^F]FDG-positive nodes, 380 (88%) received interventions targeting these lymph nodes: 84% of these 380 patients received nodal boosting, 78% extended-field radiotherapy, and 12% debulking surgery. [^18^F]FDG-positive nodes in patients receiving these treatments were more likely to be classified as suspicious than inconclusive (*p* = 0.009), located in the para-aortic region (*p* < 0.001), and larger (*p* < 0.001) than in patients who did not receive these treatments. Conclusion: While existing guidelines advocate [^18^F]FDG-PET/CT-guided treatment planning for the management of advanced cervical cancer, this study highlights that not all cases of [^18^F]FDG-positive nodes received an intervention, possibly due to the risk of false-positive results. Improvement of nodal staging may reduce suboptimal treatment planning.

## 1. Introduction

Cervical cancer is one of the most common cancers in women worldwide, with approximately 604,000 new cases and 342,000 deaths in 2020 [1]. Around 40% of cervical cancer patients are diagnosed with locally advanced disease, defined as International Federation of Gynecology and Obstetrics (FIGO) 2018 stage IB3, IIA2-IVA or FIGO 2009 stage IB2, IIA2-IVA [2,3,4]. In this group, the five-year overall survival rate after completion of standard treatment with chemoradiotherapy is ~66% [5]. Survival is worse in patients with lymph node metastases, especially in the para-aortic region [6,7]. Based on one of the few studies with prospective data from a relatively large cohort (*n* = 120), the prevalence of pathologically confirmed pelvic and para-aortic metastases in locally advanced cervical cancer is 51% and 24%, respectively [8]. This group of patients may benefit from nodal therapy in addition to primary chemoradiotherapy by boosting, extended-field radiotherapy, or debulking [9,10,11,12,13,14,15,16,17].

Guidelines recommend the use of 2-deoxy-2-[^18^F]fluoro-D-glucose positron emission computed tomography ([^18^F]FDG-PET/CT) to assess lymph node metastases and guide treatment planning in locally advanced cervical cancer [18,19]. If a suspicious lymph node is detected, an additional radiation boost should be applied and debulking may be considered, whereas metastases confined to the para-aortic region should be treated with extended-field radiotherapy [18]. [^18^F]FDG-PET/CT detects increased glucose metabolism, a characteristic of tumour cells. Detection of FDG uptake helps to differentiate between physiologically enlarged lymph nodes and metastatic nodes. It also facilitates the identification of smaller metastases compared with conventional imaging, but at the expense of detecting false-positive reactive nodes [20].

A recent meta-analysis reported positive-predictive values of 68–96% for detecting pelvic and/or para-aortic metastases in locally advanced cervical cancer, depending on the prevalence of lymph node metastases (15–65%) [21]. In other words, [^18^F]FDG-positive nodes may be false-positive in up to one-third of these patients. Therapy-related adverse events, such as surgical complications from nodal debulking and genitourinary and gastrointestinal toxicity from radiotherapy, require caution in treatment planning [10,22,23]. On the other hand, inadequately treated [^18^F]FDG-positive nodes representing true metastases could reduce the chance of survival.

Despite this daily dilemma in clinical practice, only a few studies have assessed the management of [^18^F]FDG-positive nodes with nodal boosting, extended-field radiotherapy, and nodal debulking. Therefore, this study evaluates how often patients with advanced-stage cervical cancer and an [^18^F]FDG-positive lymph node receive nodal boosting, extended-field radiotherapy, and/or debulking in addition to standard field/dose primary chemoradiotherapy.

## 2. Materials and Methods

### 2.1. Study Design and Data Collection

For this retrospective study, all cervical cancer patients diagnosed in 2009–2017, with suspected pelvic and/or para-aortic lymph node metastases on [^18^F]FDG-PET/CT, were selected from the population-based Netherlands Cancer Registry (NCR). Patients with (1) an age of ≥18 years, (2) International Federation of Gynaecology and Obstetrics (FIGO) 2009 stage IB2, IIA2-IVA, and (3) primary chemoradiotherapy were included. Patients were excluded if they had (1) a previous malignancy or a concurrent malignancy interfering with cervical cancer therapy, (2) pathological examination of suspicious nodes before nodal debulking or primary therapy, (3) a pregnancy during cervical cancer treatment, or (4) neoadjuvant chemotherapy.

Patient, tumour, imaging, and treatment characteristics were recorded retrospectively from patient records by trained data managers. Lymph node status on [^18^F]FDG-PET/CT was registered for five anatomic regions: pelvic left/right, common iliac left/right (including presacral nodes), and para-aortic, conforming with research by Liu et al. (2016) [24]. They were recorded as negative, inconclusive, suspicious, or unknown, as reported by the nuclear medicine physician. A lymph node was considered suspicious if recorded as inconclusive or suspicious, which would normally include nodes with a short-axis diameter of ≥1.0 cm and/or focally increased FDG uptake (more than the adjacent vessel), as imaging was performed according to local protocols following Dutch (Nedpas) and international (EARL) standards [25].

The management of cervical cancer in The Netherlands is based on European treatment guidelines, with limited local variation [18,19]. According to these guidelines, chemoradiotherapy consists of pelvic external beam radiotherapy (i.e., 45–50 Gy) and concurrent chemotherapy (i.e., cisplatin 40 mg/m^2^ weekly) or hyperthermia. Additional treatment of [^18^F]FDG-positive lymph nodes includes boosting, extended-field radiotherapy, and/or debulking. Debulking, or surgical resection, addresses bulky nodes without a definitive size specification [19]. Moreover, patients can receive nodal boosting for [^18^F]FDG-positive nodes with a targeted higher dose of radiation. The predetermined total dose for a nodal boost, including the contribution of brachytherapy, is 55 to 60 Gy (equi-effective dose to 2 Gy per fraction (EQD2), assuming an α/β of 10 Gy for tumours). In addition, in accordance with the EMBRACE protocol, radiotherapy can be extended to the para-aortic region in cases with common iliac or para-aortic involvement [26].

### 2.2. Outcomes and Definitions

The primary outcome of the study was the overall treatment rate of [^18^F]FDG-positive nodes in addition to standard chemoradiotherapy and for each nodal treatment separately. Nodal treatment included: (1) boost irradiation, (2) extended-field radiotherapy for common-iliac and/or para-aortic involvement, and (3) debulking ± lymphadenectomy, combined with primary chemoradiotherapy. Patients who had lymph node debulking were excluded from analysis on nodal boosting and extended field radiotherapy, but not vice versa; debulking may have been followed by nodal boosting and/or extended-field radiotherapy. For each nodal treatment strategy (i.e., boosting, extended-field radiotherapy, and/or debulking), baseline characteristics were compared between patients who did and did not receive the treatment, to identify factors that may have influenced treatment decisions. Subgroup analyses were performed to assess the impact of para-aortic [^18^F]FDG-positive nodes on extended-field radiotherapy rates, and to assess the impact of bulky nodes (with a short-axis of ≥15 mm) on nodal debulking rates. Overall survival was defined as the interval from diagnosis to death. Patient vital status was obtained by linkage to the Municipal Personal Records Database (updated to 31 January 2023). Patients who were still alive were censored at that time.

### 2.3. Statistical Analysis

The rate of patients receiving nodal treatment for [^18^F]FDG-positive nodes was calculated by dividing the number of patients receiving nodal treatment by all patients with [^18^F]FDG-positive nodes. Normally and non-normally distributed variables were compared using unpaired *t*-tests and Mann–Whitney U tests, respectively. Discrete variables were compared using Fisher’s exact tests. Survival analyses were performed using the Kaplan–Meier method. A *p*-value < 0.05 was considered significant, and Stata™ statistical software version 17.0 (StataCorp, College Station, TX, USA) was used for all analyses.

## 3. Results

In total, 434 patients with locally advanced cervical cancer and at least one [^18^F]FDG-positive lymph node on pretreatment [^18^F]FDG-PET/CT were included (see Figure 1).

In 88% of these patients (380/434), the [^18^F]FDG PET/CT lymph node information was used for additional treatment of the lymph nodes, as shown in Table 1.

Baseline characteristics of patients with and without treatment with nodal boosting, extended-field radiotherapy, and/or debulking are shown in Table 2. [^18^F]FDG-positive nodes in patients receiving these treatments were more likely to be suspicious (95% versus 85%; *p* = 0.009), located in the para-aortic region (23% versus 0%; *p* < 0.001), and larger (median short-axis of 13 mm versus 10 mm; *p* < 0.001) than in patients who did not receive these treatments.

Nodal boosting, extended-field radiotherapy, or debulking separately was observed in 84%, 78%, and 12% of patients, respectively. Nodal debulking was followed by boost and/or extended-field radiotherapy in 75% (*n* = 39). Boost with extended-field radiotherapy (without debulking) was given to 29% of patients (*n* = 109). After debulking, 2/52 patients (4%) were pathologically negative for metastasis. The 5-year overall survival rates after boosting, extended-field radiotherapy, and debulking were 67% (95% confidence interval 61–72%), 49% (38–59%), and 53% (39–66%), respectively. Baseline characteristics stratified by treatment modality are shown in Appendix A. Notably, the boosting group had larger [^18^F]FDG-positive nodes than the group without boosting (12 mm vs. 10 mm; *p* = 0.02). Patients with extended-field radiotherapy were more likely to have para-aortic involvement compared with patients who received pelvic radiotherapy (74% versus 17%; *p* < 0.001). In addition, patients treated with nodal debulking had larger tumours (55 mm versus 50 mm; *p* = 0.017), larger [^18^F]FDG-positive nodes (21 mm versus 12 mm; *p* < 0.001), and more often para-aortic involvement (37% versus 18%; *p* = 0.003). Subgroup analyses of patients with bulky node(s) (≥15 mm) increased the rate of nodal treatment by debulking from 12% to 33%. In addition, analysis of patients with para-aortic involvement increased the rate of extended-field radiotherapy from 78% to 94%.

**Table 2 cancers-16-00717-t002:** Baseline characteristics of patients receiving nodal treatment and of those who did not.

		Nodal Treatment				
Baseline Characteristics	Missing	Without (*n* = 54)		With (*n* = 380)		*p*-Value
Median age, years	0	50	(26–88)	49	(22–82)	0.17
Median body mass index, kg/m^2^	23	26	(15–36)	24	(15–77)	0.20
Charlson Comorbidity Index	70					
0		37	82.0%	252	79.0%	0.95
1		7	15.6%	53	16.6%	
≧2		1	2.2%	14	4.4%	
FIGO 2009 stage	0					
IB2		3	5.6%	56	14.7%	0.20
IIA2		1	1.9%	16	4.2%	
IIB		27	50.0%	197	51.8%	
IIIA		3	5.5%	12	3.2%	
IIIB		16	29.6%	75	19.7%	
IVA		4	7.4%	24	6.3%	
Median tumour size, mm	21	50	(24–220)	50	(20–105)	0.70
Histological subtype	0					
Squamous cell carcinoma		46	85.2%	336	88.4%	0.61
Adeno(squamous) carcinoma		7	13.0%	37	9.7%	
Other carcinomas		1	1.9%	7	1.8%	
Additional imaging techniques	0					
CT		11	20.4%	109	28.7%	0.26
MRI		49	90.7%	356	93.7%	0.39
Status of [^18^F]FDG-positive node	0					
Suspicious		46	85.2%	362	95.3%	0.009 *
Inconclusive		8	14.8%	18	4.7%	
FDG-positive nodes per region ^1^						
Pelvic	1	53	98.2%	373	98.2%	1.00
Common iliac	6	4	7.4%	69	18.2%	0.51
Para-aortic	6	0	0.0%	86	22.6%	<0.001 *
Median short axis of suspicious node, mm	85	10	(6–26)	13	(6–86)	<0.001 *

Data represent the number of patients, percentages, or median with (range). ^1^ Patients may have positive lymph nodes in multiple regions, * statistically significant. Abbreviations: FIGO, International Federation of Gynaecology and Obstetrics; CT, computed tomography; MRI, magnetic resonance imaging; FDG, fluoro-D-glucose.

## 4. Discussion

This study showed that treatment strategies was guided by [^18^F]FDG-PET/CT in 88% of patients with advanced-stage cervical cancer and [^18^F]FDG-positive nodes. Among these strategies, nodal boosting was the predominant intervention (84%) for managing [^18^F]FDG-positive nodes, followed by extended-field radiotherapy (78%) and debulking (12%). Despite existing guidelines advocating [^18^F]FDG-PET/CT-guided treatment planning for the management of advanced cervical cancer, this study highlights that not all cases of [^18^F]FDG-positive nodes received an intervention. This raises the question of whether these patients were undertreated or additional treatment was intentionally withheld to prevent overtreatment.

Undertreatment of lymph node metastases can reduce survival, and should therefore be minimised. While in 88% of the patients with [^18^F]FDG-positive nodes the treatment policy was according to the current guidelines, the remaining 12%, not following the guidelines, could theoretically have been undertreated. For nodal boosting and/or nodal debulking, there is no level 1 evidence that these treatment strategies result in better oncological outcomes [18,27,28,29,30]. Furthermore, there is no proven superiority for either boosting or debulking, nor in the context of bulky nodes (short axis ≥ 1.5 cm) [31]. Therefore, current guidelines consider, rather than recommend, these treatments for suspicious nodes [18,19]. However, several studies, including randomized controlled trials, have shown a survival benefit after extended-field radiotherapy for suspicious common-iliac/para-aortic nodes [15,16,17]. In our study, 78% of patients received extended-field radiotherapy, resulting in potential undertreatment in 22% of patients, which is relatively high compared with other reports (0–27%) [32,33,34]. This may be related to the proportion of inconclusive nodes and the portion of presacral nodes which were registered as common iliac nodes in our study. This hypothesis is supported by our analysis of patients with para-aortic involvement only, of whom 94% received extended-field radiotherapy.

On the other hand, overtreatment is a serious concern because of potential therapy-related toxicity. Unacceptable high acute (27–81%) and late (17–40%) grade ≥ 3 toxicity rates, mostly gastrointestinal and genitourinary, have been reported for conventional radiotherapy techniques [35,36]. Fortunately, improved techniques (e.g., intensity-modulated radiotherapy) have reduced toxicity rates to 4–41% and 3–29%, respectively [9,13,14,16,17,29,37]. Overtreatment is caused by targeting false [^18^F]FDG-positive nodes. According to a recently published systematic review including 778 patients with locally advanced cervical cancer, the positive predictive value of [^18^F]FDG-PET/CT varies from 68% to 96%, depending on the prevalence of lymph node metastases (range 15–65%) [21].

Using prevalences of pathologically confirmed pelvic (51%) and para-aortic (24%) lymph node metastases from a prospective, comparable study cohort, together with corresponding positive predictive values of 93% and 65%, respectively, for [^18^F]FDG-positive nodes from a meta-analysis [9,21], overtreatment of pelvic nodes with boosting and para-aortic nodes with extended-field (chemo)radiotherapy in 7% and 35% of patients, respectively, may occur. In our study population, this would have resulted in overtreatment with boosting in 22/382 (6%) patients and with extended-field radiotherapy in 30/110 (27%) patients. Therefore, caution should be exercised, especially in the group of patients where extended-field (chemo)radiotherapy is considered only on the basis of [^18^F]FDG-PET/CT. In these circumstances, other variables that increase the likelihood of nodal metastases, such as FIGO stage, larger tumour size, parametrial invasion, and the presence of lymphovascular space invasion, should also be taken into account. In the end, fine-needle aspiration or debulking of [^18^F]FDG-positive nodes is the only strategy that could potentially reduce radiotherapy-related toxicity and overtreatment, by reducing the required doses and adjusting the radiotherapy field settings. However, nodal debulking is associated with surgical complications, such as infection and intraoperative injury, with a prevalence of 10–15% [10,27].

Two limitations of this study need to be addressed. First, because of its retrospective nature with limited data, we do not know the reason why [^18^F]FDG-positive lymph nodes were not treated, which may have biased our results. In The Netherlands, most treatment recommendations are made at a multidisciplinary meeting, where patient and physician preferences, but also MRI/CT imaging results, may influence treatment decisions. In addition, [^18^F]FDG-positive nodes may be reassessed and sometimes reclassified as not suspicious, which may not have been reported accurately. These cases may have negatively affected our nodal treatment rates. Second, the characteristics of the [^18^F]FDG-positive nodes that were treated differed from those that were not treated in terms of status, location, and size. The untreated nodes appeared less suspicious and may not have been treated for this reason. Despite these limitations, this study provides insight into the implementation of current guidelines with an appraisal of its consequences for the treatment of [^18^F]FDG-positive nodes with boosting, extended-field radiotherapy, and debulking, based on a relatively large patient cohort (*n* = 434).

Although [^18^F]FDG-PET/CT is currently considered the best imaging modality for assessing nodal status in patients with locally advanced cervical cancer, suboptimal negative and positive predictive values (especially for para-aortic nodes) may still lead to inadequate treatment planning. For future research, it would be relevant to improve this value (e.g., by artificial intelligence or nomograms) or to determine treatment rates after the pathologic verification of [^18^F]FDG-positive nodes (e.g., surgical staging or imaging-guided biopsy). Currently, the PARa-aOrtic LymphAdenectomy in locally advanced cervical cancer (PAROLA)-trial is open for recruitment, investigating the effect of para-aortic surgical staging on treatment modification and recurrences in patients with suspicious pelvic nodes [38]. Furthermore, to improve the understanding of the overall efficacy and safety of the nodal treatment strategies explored in this study, it may be interesting for future research to further investigate survival outcomes and complication rates beyond prevalence.

## 5. Conclusions

In conclusion, treatment planning based on [^18^F]FDG-PET/CT was applied in 88% of patients with locally advanced cervical cancer and [^18^F]FDG-positive lymph nodes, mainly consisting of nodal boosting (84%) followed by extended-field radiotherapy (78%) and debulking (12%). Nodal treatment for [^18^F]FDG-positive lymph nodes should be weighed and discussed for each individual patient in terms of the risk of false-positivity/negativity, morbidity, and survival benefit. Future research may reduce suboptimal treatment planning by improving nodal staging.

## Figures and Tables

**Figure 1 cancers-16-00717-f001:**
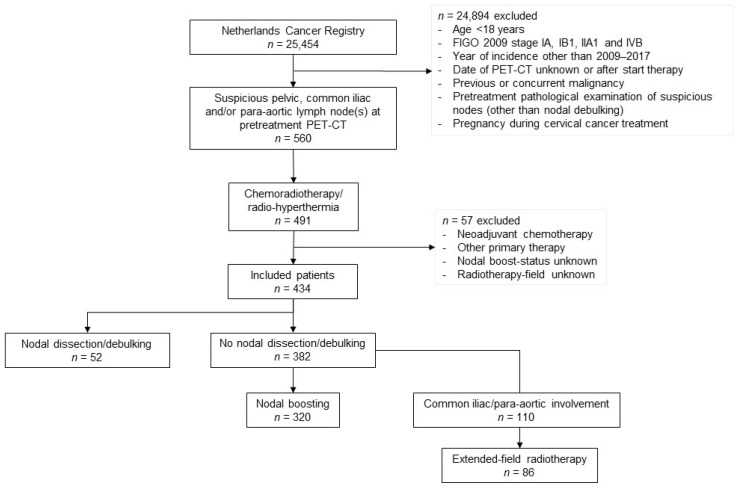
Flowchart of patient inclusion and exclusion in this study.

**Table 1 cancers-16-00717-t001:** [^18^F]FDG-positive nodal treatment rates.

		Overall	Nodal Boosting	Extended-Field ^1^		Nodal Debulking ^2^	
[^18^F]FDG-positive nodal treatment	*n*	380/434	320/382	86/110	63/67	52/434	42/127
	%	88	84	78	94	12	33

Data represent the number of patients (*n*) or percentages (%). ^1^ The right row concerns patients with [18F]FDG-positive para-aortic nodes. ^2^ The right row concerns patients with nodes ≥15 mm. Abbreviations: FDG, fluoro-D-glucose.

## Data Availability

The original contributions presented in the study are included in the article, further inquiries can be directed to the corresponding author.

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
