# Peer review of "Treatment Strategies Guided by [18F]FDG-PET/CT in Patients with Locally Advanced Cervical Cancer and [18F]FDG-Positive Lymph Nodes"

_cancers, 2024, doi:10.3390/cancers16040717_

Round 1
Reviewer 1 Report
Comments and Suggestions for Authors
This study described treatment strategies guided by [18F]FDG-PET/CT in patients with locally advanced cervical cancer and [18]FDG-positive lymph nodes.
Major comments
This study only described prevalence of three treatment strategies (nodal boosting, extended-field radiotherapy and debulking). The authors should present further information, such as survival outcomes and complication rates, in each treatment strategies.
Comments on the Quality of English LanguageMinor editing of English language required.
Author Response
Dear reviewer,
We would like to express our sincere thanks for taking the time to review our manuscript and for providing valuable suggestions and insights. Your constructive feedback has been instrumental in improving the quality of our paper. We have carefully addressed your comments, see the uploaded document.
We hope it meets your expectations.
On behalf of all the authors,
Best regards,
Ester P. Olthof, PhD Candidate
Research & Development Department
Dutch Comprehensive Cancer Organisation (IKNL)

Reviewer 2 Report
Comments and Suggestions for Authors
This original study explores treatment strategies guided by [18F]FDG-PET/CT in patients with locally advanced cervical cancer and [18F]FDG-positive lymph nodes.
The topic of this study is very interesting.
I would like to congratulte the authors as each section of the article is well-written.
This article also provides useful information to the readers of the journal about the management of locally advanced cervical cancer evaluated by [18F]FDG PET/CT.
Author Response
Dear reviewer,
We would like to thank you for taking the time to review our manuscript and for your generous comments.
On behalf of all the authors,
With kind regards,
Ester P. Olthof, PhD Candidate
Research and Development Department
Netherlands Comprehensive Cancer Organisation (IKNL)

Reviewer 3 Report
Comments and Suggestions for Authors
I have the following comments on this interesting manuscript:
1) Abstract Results (lines 37-40). Please add the results of the inferential statistical analysis illustrated in the Results section of the manuscript body, along with p-values of the comparisons between the various groups. This is important to show which findings were statistically significant and draw readers' attention to the most important ones.
2) Abstract Conclusions (lines 41-42). Please add a sentence summarizing the actions that should be taken to address the issues related to the fact that despite current guidelines, not all cases of [18F]FDG-positive nodes received an intervention.
3) Materials and methods (lines 97-98). It appears that each nuclear medicine physician's report was given for granted and used as ground truth, which may introduce a bias because the lymph node classification used can be quite subjective (e.g., the ranking "suspicious" could lead to a non-negligible degree of uncertainty). Therefore, it would be ideal to have all available PET-CT examinations rechecked in blind by at least two different nuclear medicine physicians, so as to verify the lymph node assessment and quantify inter-rater agreement. Otherwise, the aforementioned circumstance should be acknowledged as an additional study limitation in the Discussion section.
4) Materials and methods (line 112). The expression "assuming an / of 10Gy for tumour" contains bad characters and should be modified to make it understandable.
5) Discussion (lines 202-204). Please explain the reasons for this misclassification of lymph nodes, and again (see comment #3), consider having PET-CT examinations rechecked also in order to address this issue.
Author Response
Dear reviewer,
We would like to express our sincere thanks for taking the time to review our manuscript and for providing valuable suggestions and insights. Your constructive feedback has been instrumental in improving the quality of our paper. We have carefully addressed your comments, see the uploaded document.
We hope it meets your expectations.
On behalf of all the authors,
With kind regards,
Ester P. Olthof, PhD Candidate
Research & Development Department
Netherlands Comprehensive Cancer Organisation (IKNL)

Round 2
Reviewer 1 Report
Comments and Suggestions for Authors
Thank you for your efforts to revise manuscript.
Author Response
Dear reviewer,
We would like to thank you again for taking the time to review our manuscript and for your suggestions. In response to your comment, we have included the requested survival data, see the uploaded document.
We hope this meets your expectations.
On behalf of all the authors,
With kind regards,
Ester P. Olthof, PhD Candidate
Research & Development Department
Netherlands Comprehensive Cancer Organisation (IKNL)

Reviewer 3 Report
Comments and Suggestions for Authors
Thank you. No further comments.
Author Response
We appreciate your agreement with our revisions and the considerable time and effort you have devoted to evaluating our manuscript. Thank you for your invaluable input.